# Robustness between the worst and average case

**Leslie Rice**
Department of Computer Science
Carnegie Mellon University
Pittsburgh, PA
`larice@cs.cmu.edu`

**Anna Bair**
Department of Machine Learning
Carnegie Mellon University
Pittsburgh, PA
`abair@cmu.edu`

**Huan Zhang**
Department of Computer Science
Carnegie Mellon University
Pittsburgh, PA
`huan@huan-zhang.com`

**J. Zico Kolter**
Department of Computer Science
Carnegie Mellon University &
Bosch Center for Artificial Intelligence
Pittsburgh, PA
`zkolter@cs.cmu.edu`

## Abstract

Several recent works in machine learning have focused on evaluating the test-time robustness of a classifier: how well the classifier performs not just on the target domain it was trained upon, but upon perturbed examples. In these settings, the focus has largely been on two extremes of robustness: the robustness to perturbations drawn *at random* from within some distribution (i.e., robustness to random perturbations), and the robustness to the *worst case* perturbation in some set (i.e., adversarial robustness). In this paper, we argue that a sliding scale between these two extremes provides a valuable additional metric by which to gauge robustness. Specifically, we illustrate that each of these two extremes is naturally characterized by a (functional) $q$-norm over perturbation space, with $q = 1$ corresponding to robustness to random perturbations and $q = \infty$ corresponding to adversarial perturbations. We then present the main technical contribution of our paper: a method for efficiently estimating the value of these norms by interpreting them as the partition function of a particular distribution, then using path sampling with MCMC methods to estimate this partition function (either traditional Metropolis-Hastings for non-differentiable perturbations, or Hamiltonian Monte Carlo for differentiable perturbations). We show that our approach provides substantially better estimates than simple random sampling of the actual "intermediate-$q$" robustness of standard, data-augmented, and adversarially-trained classifiers, illustrating a clear tradeoff between classifiers that optimize different metrics. Code for reproducing experiments can be found at `https://github.com/locuslab/intermediate_robustness`.

## 1 Introduction

There has been an increasing focus in recent years in evaluating the *robustness* of machine learning classifiers, broadly interpreted as evaluating their performance not just on a test set, but also evaluating the performance relative to some additional (possibly domain-specific) uncertainty or bounds on the problems. Although there are many formal definitions of robustness, most work in this area has focused on two particular settings. In the "classical" sense of robustness, we can consider evaluating the classifier in terms of its *worst-case* loss under some perturbation set applied to the inputs, i.e., we

35th Conference on Neural Information Processing Systems (NeurIPS 2021).

could evaluate (via finite sample approximation)

$$\mathbf{E}_{x,y\sim\mathcal{D}}\left[\max_{\delta\in\Delta(x)}\ell(h(x+\delta),y)\right] \tag{1}$$

where $\mathcal{D}$ denotes a distribution over $x, y$ pairs, $h$ denotes the hypothesis function, $\ell$ denotes a loss, and $\Delta(x)$ denotes some (input-dependent) *uncertainty region*. This formulation, for instance, underlies adversarial examples and also motivates the classical adversarial training approaches. However, substantial work has also been done in evaluating the setting of robustness to *random* perturbations, i.e., evaluating a classifier via the loss

$$\mathbf{E}_{x,y\sim\mathcal{D}}\left[\mathbf{E}_{\delta\sim\mathcal{P}(x)}[\ell(h(x+\delta),y)]\right] \tag{2}$$

where now $\mathcal{P}(x)$ denotes some (again, input-dependent) *distribution* over possible perturbations. This formulation underlies common data augmentation strategies in deep learning, as well as most formulations of "natural" robustness [Hendrycks and Dietterich, 2018] (even if not always written in this formal manner).

Until now, these two types of robustness have typically been seen as largely separate notions. We believe there is inherent value in generalizing these two notions to place them in a unified framework. The main criticism of worst-case robustness is that it focuses "too much" on the worst case, while the criticism of average-case is that it is "not robust enough". It seems very likely that what people actually want in terms of robustness is precisely something in the middle, between these two extremes.

In this paper, we advocate for a more fine-grained spectrum of robustness definitions, which naturally interpolates between both these two extremes. In particular, we argue that robustness to random perturbations and worst-case robustness can be naturally interpreted as (functional) $\ell_q$ norms[1] of the loss function evaluated over the perturbation distribution (which can be a uniform distribution in the case of traditional adversarial loss). In particular, the random setting corresponds to the choice of $q = 1$ and the adversarial setting corresponds to the choice of $q = \infty$. We believe that it is also valuable and informative to consider the performance of classifiers in a wide range in between these two extremes, i.e., the performance of "intermediate-$q$" robustness. However, evaluating this intermediate-$q$ robustness is non-trivial, owing to the fact that it requires computing a high dimensional integral over the perturbation space. Thus, our main technical contribution of this paper is the proposal of a simple approach to evaluating the relevant robustness norms, using a combination of path sampling and Markov chain Monte Carlo (MCMC) approaches (specifically the Hamiltonian Monte Carlo method for cases where loss is a differentiable function of the perturbation space). Despite their seeming complexity, in this particular case the eventual estimators take the very simple form of a geometric mean computed over samples from an annealed distribution over the perturbation region.

We evaluate our approach on networks trained via standard training, data augmentation, and adversarial training. In all the cases our proposed approach shows a clear trade-off between different levels of robustness that would missed by solely considering just the random or adversarial perturbation setting. Furthermore, we show that our HMC-based path sampling estimator for intermediate-$q$ robustness is vastly superior to naive estimates produced e.g. by Monte Carlo sampling. We also briefly highlight the possibility of actually *training* networks using these estimators to create classifiers more robust to these intermediate notions of robustness. Code for reproducing experiments can be found at https://github.com/locuslab/intermediate_robustness.

## 2 Background and related work

**Robustness in deep learning** Adversarial robustness is the "classicial" view of robustness that traces its roots back to robust optimization, a field in optimization theory that allows for the study of worst-case uncertainty [Ben-Tal et al., 2009]. Szegedy et al. [2013] noted the marked presence of adversarial examples in deep classifiers, in which the addition of imperceptible amounts of noise leads to gross misclassification of input images by deep networks. Goodfellow et al. [2014] then introduced the Fast Gradient Sign Method (FGSM), which uses a single gradient step to generate adversarial

---

[1]We should emphasize that this use of $\ell_q$ norms is entirely orthogonal to the use of $\ell_p$ balls as perturbation regions, commonly done in adversarial robustness. The $\ell_q$ norms here can be applied to any perturbation region, as we will highlight.

examples and was used to perturb training examples as the original form of adversarial training. Subsequently, the Basic Iterative Method was introduced by Kurakin et al. [2016], and rendered FGSM adversarial training ineffective through attacks using multiple, smaller FGSM steps (although the robustness of FGSM adversarial training was later improved [Wong et al., 2020, Andriushchenko and Flammarion, 2020]). The work of Madry et al. [2017] further improved upon this iterative attack, and incorporated the resulting Projected Gradient Descent (PGD) adversary into adversarial training. Today, PGD remains the most empirically effective adversary for adversarial training and attacks. While traditional adversarial training focuses on perturbations within some $\ell_p$-norm ball, it has also been expanded to more realistic threat models, from spatial transformations [Engstrom et al., 2019, Xiao et al., 2018] to weather corruptions and lighting changes [Wong and Kolter, 2020].

However, when considering robustness to more "realistic" (yet synthetic) perturbations, the focus is often shifted from robustness to the worst-case perturbation within some set to robustness to random perturbations [Geirhos et al., 2018, Hendrycks and Dietterich, 2018, Hendrycks et al., 2019b,a, Yin et al., 2019]. It was been further studied whether such synthetic corruptions can universally improve robustness to real-world distribution shifts, such as geographic location, or camera hardware [Hendrycks et al., 2020]. Other naturally occurring distribution shifts of interest include consistency shifts [Shankar et al., 2019] and dataset shifts [Storkey, 2009, Taori et al., 2020]. For example, the impact of dataset shift has been emphasized on the ImageNet dataset, where the performance of models trained on ImageNet was shown to be drastically worse on a reproduction of the validation set [Recht et al., 2019]. While some work such as Meunier et al. [2021] could be interpreted as an interpolation between random and adversarial noise, generally the field faces a divide between studies on adversarial robustness and robustness to random data perturbations.

**MCMC methods for partition function estimation** Because our definition of intermediate-$q$ robustness involves computation of a high dimensional integral over a perturbation distribution, our work can also be related to methods for estimating an intractable partition function of a probability distribution. The problem of computing normalizing constants ties to the problem of computing differences in free energy in physics. Several Monte Carlo sampling-based approaches exist for approximately estimating normalizing constants. Importance sampling is one approach at estimating ratios of normalizing constants that relies on samples from one of the distributions. Bridge sampling [Bennett, 1976, Meng and Wong, 1996], on the other hand, uses a single "bridge" distribution to interpolate between the two distributions and can reduce the Monte Carlo errors associated with importance sampling. Annealed importance sampling [Jarzynski, 1997, Neal, 2001] bridges the gap between the two distributions by chaining together intermediate distributions, and linked importance sampling [Neal, 2005] then combined bridge sampling with annealed importance sampling to bridge these intermediate distributions. Path sampling [Gelman and Meng, 1998], which originated under the name of thermodynamic integration in physics [Ogata, 1989], aims at reducing Monte Carlo errors by introducing flexible paths.

Our proposed approach also leverages Hamiltonian Monte Carlo methods to improve the sample efficiency of our eventual estimator [Duane et al., 1987, Betancourt, 2017, Neal et al., 2011]. Hamiltonian Monte Carlo (HMC) is useful due to its avoidance of random walks, and is compatible with sampling in a constrained space through use of reflection [Afshar et al., 2015].

## 3 A simple generalization of robustness

To begin, we first make the simple observation that there is a natural interpolation between the notions of adversarial robustness and robustness to random perturbations. Specifically, we note that both these notions can be expressed as function $\ell_q$ norms over the perturbation space. Specifically, we define the following functional norm

**Definition 1.** *For a continuous function $f : \mathbb{R}^n \to \mathbb{R}$ and density $\mu : \mathbb{R}^n \to \mathbb{R}_+$, $\int \mu(x)dx = 1$, let $\|f\|_{\mu,q}$ be the q-norm of the function under this density*

$$\|f\|_{\mu,q} = \mathbf{E}_{x \sim \mu}\left[|f(x)|^q\right]^{1/q} = \left(\int |f(x)|^q \mu(x)dx\right)^{1/q}. \tag{3}$$

Then as the proposition shows, random perturbation loss and adversarial perturbation loss simply correspond to two extremes of this functional norm over the perturbation $\delta$, as formalized by the following proposition.

**Proposition 1.** *Let $\delta$ be a random variable with density $\mu$ and consider the expectation*

$$\mathbf{E}_{x,y\sim D}\Big[\|\ell(h(x+\delta),y)\|_{\mu,q}\Big]. \tag{4}$$

*Then (for a smooth loss $\ell$) this corresponds to the expected loss on random samples from $\mu$ when $q = 1$, and to the expected adversarial loss over the domain of $\mu$ when $q = \infty$.*

This proposition follows immediately from the fact that losses are non-negative, and the fact that the $\ell_\infty$ norm is given by the pointwise maximum of the function, assuming a smooth loss $\ell$ (i.e. not zero-one loss). Note that "traditional" adversarial loss actually arises more specifically when $q = \infty$ and $\mu$ is a uniform distribution over some norm ball (unrelated to the norm $q$). When we have $1 < q < \infty$, we enable a full spectrum of robustness measurements, which we refer to as *intermediate-$q$ robustness*, that evaluates the performance of classifiers in a wide range in between these two extreme cases. Furthermore, we argue that there are naturally appealing properties of these intermediate-$q$ robustness measures: whereas adversarial robustness may overestimate the risk of what effectively amount to "measure zero" regions of the perturbation space, random robustness may likewise fail to take into account smaller but non-negligible regions that *do* contain areas of high loss.

As an example, when considering $\delta$ as a Gaussian distribution, adversarial loss ($q = \infty$) is not meaningful because the worst case perturbation can be arbitrarily far away from $x$. Meanwhile, random data augmentation with Gaussian noise ($q = 1$) is often insufficient for evaluating robustness because random Gaussian samples are rarely good adversarial examples. Our intermediate-$q$ robustness allows us to consider a middle ground where the model is robust under a certain degree of adversarial noise (stronger than randomly sampled Gaussian noise) while the evaluation is not hindered by very rare events (e.g., an extremely low probability Gaussian sample far away from $x$).

## 4 Estimating intermediate-$q$ robustness via MCMC methods

Of course, simply writing the loss in this manner is not particularly useful on its own. In most cases, the integral in Equation 3 cannot be computed exactly, and so we must resort to numerical approximation methods. Specifically, in order to estimate the $q$-norm robustness for an arbitrary nonlinear model $h$, loss $\ell$, and (known) density $\mu$, we consider the problem of computing the integral

$$Z := \left( \int \ell(h(x+\delta),y)^q \mu(\delta)d\delta \right)^{(1/q)}. \tag{5}$$

One could naively estimate this integral by using Monte Carlo sampling, and sample $\delta^{(1)}, \ldots, \delta^{(m)}$ randomly from density $\mu$, and approximate the objective as the following,

$$\hat{Z}_{\text{Monte Carlo}} = \left( \frac{1}{m}\sum_{i=1}^{m} \ell(h(x+\delta^{(i)}),y)^q \right)^{(1/q)}. \tag{6}$$

However, because the integral in (5) will be dominated by values with large $\ell(h(x+\delta),y)$, simply using Monte Carlo sampling will be insufficient to approximate this integral well for larger values of $q$, as random sampling will place too much weight on regions of low loss.

Instead, in this work we argue that it is beneficial to interpret the task at hand as one of evaluating the *partition function* of a particular unnormalized probability density. Specifically, we define an (unnormalized) density over the perturbation $\delta$

$$\tilde{p}(\delta) = \ell(h(x+\delta),y)^q \mu(\delta). \tag{7}$$

Then clearly, just from construction, we see that the task of evaluating the ($q^{\text{th}}$ root of) partition function (the normalizing constant) of this distribution is exactly the same as that of computing the integral of interest.

The advantage of this perspective on the integral of interest, however, is that we can use a wealth of techniques developed for partition function estimation in order to better estimate this particular integral. Specifically, we argue to use the path sampling [Gelman and Meng, 1998] approach, a Markov chain Monte Carlo based method, to approximate the desired partition function. In fact, we show that for the precise form of the integral in question, the eventual estimator produced by this method takes on a very simple form: it consists of a geometric mean over samples generated from a certain annealed distribution. This makes the estimator particularly simple to implement and even in some cases to train networks based upon, via standard automatic differentiation toolkits.

## 4.1 Evaluating the particular function of the loss-based distribution

We state our main result on the path sampling formulation of the integral of interest via the following theorem.

**Theorem 1.** *Consider the task of approximately computing the following integral:*

$$Z := \left( \int \ell(h(x+\delta), y)^q \mu(\delta) d\delta \right)^{(1/q)}.$$

*Let $\{t^{(1)}, t^{(2)}, \cdots, t^{(m)}\}$ be $m$ scalars corresponding to linearly interpolated values from $0$ to $q$. For $i = 1, \ldots, m$, sample $\delta^{(i)}$ from the following unnormalized density*

$$\delta^{(i)} \sim p(\delta | t^{(i)}), \text{ where, } p(\delta | t) \propto \ell(h(x+\delta), y)^t \mu(\delta).$$

*Then the following estimator, given by the geometric mean of the resulting samples*

$$\hat{Z}_{path} := \left( \prod_{i=1}^{m} \ell(h(x+\delta^{(i)}), y) \right)^{1/m}. \tag{8}$$

*is a consistent estimator of our integral $Z$. Practically, the $m$ samples can be drawn using MCMC.*

*Proof.* For notational completeness, we define the density

$$p(\delta | t) = \frac{1}{z(t)} \tilde{p}(\delta | t) \tag{9}$$

where

$$\tilde{p}(\delta | t) = \ell(h(x+\delta), y)^t \mu(\delta). \tag{10}$$

and where $z(t)$ denotes the partition function of this distribution

$$z(t) = \int \tilde{p}(\delta | t) d\delta. \tag{11}$$

Taking the log of both sides of and differentiating with respect to $t$, we get:

$$
\begin{aligned}
\frac{d}{dt} \log(z(t)) &= \frac{d}{dt} \log \int \tilde{p}(\delta | t) d\delta \\
&= \frac{\int \frac{d}{dt} \tilde{p}(\delta | t) d\delta}{\int \tilde{p}(\delta | t) d\delta} \\
&= \int \frac{\tilde{p}(\delta | t)}{z(t)} \frac{\frac{d}{dt} \tilde{p}(\delta | t)}{\tilde{p}(\delta | t)} d\delta \\
&= \mathbf{E}_{\delta \sim p(\delta | t)} \left[ \frac{d}{dt} \log \tilde{p}(\delta | t) \right] \\
&= \mathbf{E}_{\delta \sim p(\delta | t)} \left[ \log \ell((h(x+\delta), y)) \right]
\end{aligned} \tag{12}
$$

where we use the fact that

$$\log \tilde{p}(\delta | t) = t \log \ell((h(x+\delta), y)). \tag{13}$$

Then we can integrate (12) from $0$ to $q$ to get

$$\log \left[ \frac{z(q)}{z(0)} \right] = \int_0^q \mathbf{E}_{\delta \sim p(\delta | t)} \left[ \log(\ell(h(x+\delta), y)) \right] dt \tag{14}$$

Given in our case $z(0) = 1$ and incorporating the exponent of $(1/q)$, we have

$$\log \left[ z(q)^{(1/q)} \right] = \frac{1}{q} \int_0^q \mathbf{E}_{\delta \sim p(\delta | t)} \left[ \log(\ell(h(x+\delta), y)) \right] dt \tag{15}$$

Then the right hand side of Equation 15 can be interpreted as the expectation of $\log(\ell(h(x+\delta), y))$ over the joint distribution of $(\delta, t)$, where $t$ is a random variable with a uniform distribution in $[0, q]$.

$$\log \left[ z(q)^{(1/q)} \right] = \mathbf{E}_{t \sim U[0,q]} \left[ \mathbf{E}_{\delta \sim p(\delta | t)} \left[ \log(\ell(h(x+\delta), y)) \right] \right] \tag{16}$$

**Algorithm 1** Evaluating the intermediate-$q$ robustness of a neural network function $h$ using path sampling estimation with $m$ MCMC samples with $x, y \sim D$ for some norm $q$.

---

Initialize $\delta^{(0)}$ randomly
**for** $i = 1 \ldots m$ **do**
    Let $t^{(i)} := q \cdot \frac{i-1}{m-1}$
    Sample $\delta^{(i)} \sim p(\delta | t^{(i)})$ using MCMC from initial state $\delta^{(i-1)}$
**end for**
return $\left( \prod_{i=1}^{m} \ell(h(x + \delta^{(i)}), y) \right)^{1/m}$

---

Finally, sampling $(\delta^{(i)}, t^{(i)})$ for $i = 1, \ldots, m$ from this joint distribution $p(\delta, t)$ (which we can do by linearly interpolating $t^{(i)}$ between 0 and $q$ and then sampling $\delta^{(i)}$ from $p(\delta | t^{(i)})$), we have the fact that

$$\hat{Z}_{\text{path}} := \exp \left( \frac{1}{m} \sum_{i=1}^{m} \log(\ell(h(x + \delta^{(i)}), y) \right) = \left( \prod_{i=1}^{m} \ell(h(x + \delta^{(i)}), y) \right)^{1/m}, \qquad (17)$$

is a consistent estimator of the desired integral. $\qquad\qquad\square$

The key point of this result is that it allows us to approximate the desired integral just through the ability to *sample* from the distribution $p(\delta | t)$. While this is still a challenging task, sampling from unnormalized probability distributions is a well-studied problem, and we can apply MCMC sampling methods to this task. Further, while the sampling of $(\delta^{(i)}, t^{(i)})$ can be done in different ways, we choose to linearly anneal $t^{(i)}$ from 0 to $q$, and then draw $\delta^{(i)} \sim \tilde{p}(\delta | t)$ using some MCMC sampler. This has the nice feature that it starts with sampling from an "easy" distribution (when $t = 0$, the distribution over $\delta$ is simply given by $\mu$), and gradually anneals to a more peaked distribution as $t$ increases. The resulting algorithm for evaluating a network using this geometric mean estimator is shown in Algorithm 1.

## 4.2 Sampling from the (unnormalized) loss distribution via HMC

In order to generate the samples for the path sampling estimation (17) from the desired distribution $p(\delta, t)$, we use Markov chain Monte Carlo (MCMC) methods to sample $\delta$ from the unnormalized distribution $\tilde{p}(\delta | t)$. When the loss is a differentiable function of the perturbation distribution, we can take advantage of gradient-based methods to reduce random walk behavior in MCMC sampling and achieve more efficient sampling. Hamiltonian Monte Carlo (HMC) is one such gradient-based MCMC method that simulates Hamiltonian dynamics to improve sample efficiency in high-dimensional spaces. HMC is based on the Hamiltonian function $H(q, p) = U(q) + K(p)$, where $q$ is a $d$-dimensional position vector, $p$ is a $d$-dimensional momentum vector, $U(q)$ is the potential energy, and $K(p)$ is the kinetic energy. To translate this to our setting, $U(\delta) = -\log(\ell(x + \delta), y)^t \mu(\delta))$ is just the negative log probability density of the distribution we want to sample from, and $K(p) = ||p||^2 / (2\sigma^2)$ is the negative log probability density of the zero-mean Gaussian distribution with variance $\sigma^2$. Then the Hamiltonian function $H$ is equal to the following:

$$H(\delta, p) = -t \log(\ell(h(x + \delta), y)) + \log \mu(\delta) + \frac{||p||^2}{\sigma^2} \qquad (18)$$

In order to make use of Hamiltonian dynamics in practice, Hamiltonian's equations must be discretized. The Leapfrog method is one such discretization of Hamiltonian dynamics, using a small stepsize $\alpha$ to discretize time and numerically integrating the system of differential equations as follows:

$$\begin{aligned} p &= p + \alpha t \nabla_\delta \log(\ell(h(x + \delta), y))/2 \\ \delta &= \delta + \alpha p / \sigma^2 \\ p &= p + \alpha t \nabla_\delta \log(\ell(h(x + \delta), y))/2 \end{aligned} \qquad (19)$$

The leapfrog method is reversible by simply negating $p$ (although this need not be done in practice, since $K(p) = K(-p)$). The HMC algorithm begins by first sampling a new momentum vector

$p \sim \mathcal{N}(0, \sigma^2)$, independent of the current state of $\delta$. Then, the Leapfrog method in (19) is repeated for $L$ steps to propose a new Markov state $(\delta', p')$. This proposed state is then accepted with probability $\min[1, \exp(-\Delta H)]$, where $\Delta H = H(\delta', p') - H(\delta, p)$. If the state $(\delta', p')$ is not accepted, the next state is then set to $(\delta, p)$.

Given that in general, our perturbation distribution will likely be constrained in some manner, we need to modify the HMC algorithm so that our proposals remain within the boundaries of the perturbation distribution. Consider the case where our perturbation distribution is the $\ell_\infty$ norm ball with radius $\epsilon$, so that each element of $\delta$ is constrained to be within $-\epsilon$ and $\epsilon$. In order to enforce these constraints while preserving the dynamics, we incorporate what is known as *reflection* in HMC. In this case, after setting $\delta = \delta + \epsilon p / \sigma^2$, we check if any $\delta'_i > \epsilon$ or $\delta'_i < -\epsilon$ for $i = 1, \ldots, n$. If so, we negate the corresponding momentum term, $p_i$, and if $\delta'_i > \epsilon$, we set $\delta_i = 2\epsilon - \delta'_i$, whereas if $\delta'_i < -\epsilon$, we set $\delta'_i = -2\epsilon - \delta'_i$. We repeat this reflection step until $\delta_i$ satisfies our constraints. One can think of this behavior as effectively simulating reflecting off a physical boundary.

We note that we can still easily use the path sampling estimator for non-differentiable perturbations by replacing the Hamilton Monte Carlo sampler with any other non-gradient based MCMC sampler, such as a random walk Metropolis. This is roughly similar to random sampling, but with a more subtle weighting on terms that involve higher loss.

### 4.3 Estimating the partition function during training

We further consider the possibility of *training* networks using the estimators we have discussed to achieve better intermediate robustness. However, this becomes less computationally feasible due to the number of samples required to get a good estimate of the objective, due to the $m \times L$ iterations of the path sampling estimator with Hamiltonian Monte Carlo, where $L$ is the number of Leapfrog steps. Additionally, the step size $\alpha$ and variance $\sigma^2$ in HMC require careful tuning. However, for larger values of $q$, the path sampling estimator is essential to getting accurate estimates of the training objective, as random sampling will be much less likely to come across regions of the perturbation distribution with high loss. Path sampling draws samples from the unnormalized loss distribution using MCMC, and so with increased $q$, there will be higher weighting on samples that induce higher loss. The Hamiltonian Monte Carlo method has the additional benefit of following the gradient (along with some noise), and so for larger $q$, even with a small number of iterations, path sampling can have advantages over Monte Carlo sampling during training.

## 5 Experimental evaluation

In this section, we evaluate the intermediate-$q$ robustness of models trained using standard training, data augmentation, and adversarial training, comparing estimates of the functional $q$-norm of the loss function over the perturbation distribution computed via naive, Monte Carlo sampling (6) and path sampling (8) with Hamiltonian Monte Carlo (for differentiable perturbations) and random walk Metropolis Hastings (for non-differentiable perturbations). We additionally show preliminary results of training according to the intermediate-$q$ robustness objective approximated by these estimators. All of our experiments are either run on the MNIST dataset [LeCun et al., 1998] or the CIFAR-10 dataset [Krizhevsky et al., 2009]. We consider robustness over two types of perturbations, namely the $\ell_\infty$ norm bounded input perturbation and spatial transformations. Additional results can be found in Appendix A.1 and A.2, and experimental details can be found in Appendix A.3.

### 5.1 Robustness over the $\ell_\infty$-norm ball

In order to easily translate the notion of intermediate-$q$ robustness to that of adversarial robustness, we consider perturbations uniformly distributed within an $\ell_\infty$-norm ball with radius $\epsilon$. Our evaluations of intermediate-$q$ robustness on MNIST and CIFAR-10 in Table 1 both show that estimates of the functional $q$-norm of the loss function evaluated over this perturbation distribution naturally interpolate between loss over random samples (MC $q = 1$) and adversarial loss (PGD). For larger values of $q$, specifically $q = 10^2$ and $q = 10^3$, the path sampling with HMC (PS+HMC) estimator consistently produces more accurate (i.e. higher) estimates of the desired integral than the Monte Carlo (MC) estimator. We show that this result holds for all of the models we evaluated, each of which were trained according to a different training objective, and across both datasets. The advantage of

Table 1: Evaluations comparing the Monte Carlo estimates ($\hat{Z}_{\text{MC}}$) and path sampling with HMC estimates ($\hat{Z}_{\text{PS+HMC}}$) of the functional $q$-norm of the loss over the $\ell_\infty$ ball with radius $\epsilon = 0.3$ on MNIST and $\epsilon = 0.03$ on CIFAR-10. As $q$ increases, $\hat{Z}_{\text{PS+HMC}}$ computes better estimates interpolating between random and adversarial robustness. On MNIST, $\hat{Z}_{\text{MC}}$ is computed with $m = 2000$, $\hat{Z}_{\text{PS+HMC}}$ with $m = 100, L = 20$, and Adv. loss corresponds to PGD with 100 iterations. On CIFAR-10, $\hat{Z}_{\text{MC}}$ is computed with $m = 500$, $\hat{Z}_{\text{PS+HMC}}$ with $m = 50, L = 10$, and Adv. loss corresponds to PGD with 50 iterations at 10 restarts.

| Dataset | Train method | $\hat{Z}_{\text{MC}}$ | | | | $\hat{Z}_{\text{PS+HMC}}$ | | | | Adv. loss |
|---|---|---|---|---|---|---|---|---|---|---|
| | | $q=1$ | $q=10$ | $q=10^2$ | $q=10^3$ | $q=1$ | $q=10$ | $q=10^2$ | $q=10^3$ | |
| MNIST | Standard | 0.043 | 0.140 | 0.251 | 0.268 | 0.043 | 0.160 | 1.420 | 4.456 | 11.649 |
| MNIST | MC $q=1$ | 0.032 | 0.084 | 0.143 | 0.154 | 0.032 | 0.088 | 0.692 | 2.133 | 7.363 |
| MNIST | PGD-50 | 0.039 | 0.051 | 0.076 | 0.081 | 0.039 | 0.048 | 0.101 | 0.187 | 0.270 |
| CIFAR-10 | Standard | 0.453 | 0.787 | 1.153 | 1.216 | 0.453 | 0.841 | 2.718 | 4.991 | 18.142 |
| CIFAR-10 | MC $q=1$ | 0.405 | 0.532 | 0.717 | 0.756 | 0.405 | 0.546 | 1.490 | 3.140 | 14.240 |
| CIFAR-10 | PGD-10 | 0.733 | 0.734 | 0.743 | 0.761 | 0.733 | 0.734 | 0.743 | 0.796 | 1.411 |

Table 2: Evaluation of models trained according to estimates of the functional $q$-norm of the loss over the $\ell_\infty$ ball with radius $\epsilon = 0.3$ on MNIST. As we increase $q$ during evaluation, models trained using the path sampling estimator with HMC tend to have lower intermediate-$q$ robust loss than models trained using the Monte Carlo estimator for the same $q$. Note that as we have shown in Table 1, $\hat{Z}_{\text{MC}}$ is not reliable for large $q$ so $\hat{Z}_{\text{PS+HMC}}$ should be used for comparisons. $\hat{Z}_{\text{MC}}$ is computed with $m = 2000$, $\hat{Z}_{\text{PS+HMC}}$ with $m = 100, L = 20$, and Adv. loss corresponds to PGD with 100 iterations.

| Train method | $\hat{Z}_{\text{MC}}$ | | | | $\hat{Z}_{\text{PS+HMC}}$ | | | | Adv. loss |
|---|---|---|---|---|---|---|---|---|---|
| | $q=1$ | $q=10$ | $q=10^2$ | $q=10^3$ | $q=1$ | $q=10$ | $q=10^2$ | $q=10^3$ | |
| MC $q=10$ | 0.026 | 0.058 | 0.098 | 0.105 | 0.026 | 0.058 | 0.412 | 1.336 | 3.722 |
| MC $q=10^2$ | 0.025 | 0.055 | 0.093 | 0.099 | 0.025 | 0.055 | 0.388 | 1.261 | 3.492 |
| MC $q=10^3$ | 0.025 | 0.055 | 0.093 | 0.100 | 0.025 | 0.055 | 0.390 | 1.268 | 3.488 |
| PS+HMC $q=10$ | 0.031 | 0.075 | 0.126 | 0.135 | 0.031 | 0.075 | 0.467 | 1.307 | 5.012 |
| PS+HMC $q=10^2$ | 0.028 | 0.060 | 0.099 | 0.107 | 0.028 | 0.058 | 0.304 | 0.816 | 2.613 |
| PS+HMC $q=10^3$ | 0.024 | 0.047 | 0.077 | 0.083 | 0.024 | 0.045 | 0.239 | 0.684 | 1.646 |

the PS+HMC estimator over the MC estimator is further illustrated in Figure 1, where we show the convergence of their corresponding estimates of the functional $q$-norm of the loss over this same perturbation distribution given increasing number of samples. Specifically, we compare the estimates of the intermediate-$q$ robust loss computed by each estimator given the same number of iterations (which corresponds to $m$ for the MC estimator, and $m \times L$ for the PS+HMC estimator). While for $q = 1$ the estimates converge to the same value given a similar number of iterations, for $q = 100$, the estimates quickly diverge, with path sampling producing a much higher (i.e. better) estimate.

In addition to evaluating intermediate-$q$ robustness using both estimators, we additionally show the potential benefits of training using these estimators in Table 2. On MNIST, we compare the results of training using the MC objective vs. the PS+HMC objective. We note that when $q = 1$, the MC estimate corresponds to just applying $m$ random perturbations per example and taking the average loss, similar to data augmentation methods. For the MC estimate computed during training, we use $m = 50$ samples, whereas for the PS+HMC estimate we use $m = 25$ samples with $L = 2$ leapfrog steps in order to compare an equivalent number of iterations. We find that most models trained according to the PS+HMC objective outperform those trained using the MC objective for the same value of $q$, i.e. have a lower intermediate-$q$ robust loss. This suggests that even with a reasonably small number of samples, path sampling can still result in a better estimate of the objective.

We additionally attempt training on CIFAR-10 using these estimators, however the computational complexity given the number of samples required to get a reasonable estimate makes training more challenging for larger datasets. Despite this, we see promise for training deep networks to robustness levels somewhere between robustness over random perturbations and robustness over worst-case perturbations. Full tables of results for MNIST and CIFAR-10 can be found in Tables 4 and 6 respectively in Appendix A.1.

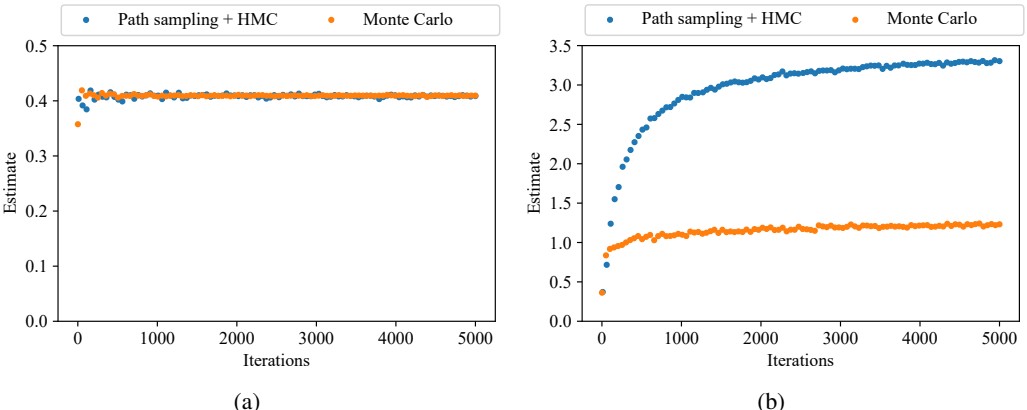

Figure 1: Comparison of the convergence of HMC-based path sampling and naive Monte Carlo estimates for (a) $q = 1$, and (b) $q = 100$ with increasing iterations on a standard trained model on CIFAR-10 using a single batch. Iterations corresponds $m$ for the Monte Carlo estimator, and to $m \times L$ for the path sampling + HMC estimator using $L = 10$ leapfrog steps. For a smaller $q = 1$ (a), random sampling works reasonably well, whereas for $q = 100$ (b), using path sampling with HMC is essential to get good estimates.

## 5.2   Robustness over (non-differentiable) spatial transformations

In order to show that the path sampling estimator can naturally be extended to non-differentiable perturbations, we consider a perturbation set consisting of parameterizations of spatial transformations on CIFAR-10. The parameters of the spatial transformations include horizontal flips, rotations between $-10$ to $10$ degrees, scaling factors between $0.9$ to $1.1$, and cropping between $0$ to $4$ pixels horizontally and vertically. Because the applied spatial perturbations are non-differentiable, in place of Hamiltonian Monte Carlo we use Gaussian random walk Metropolis Hastings to sample from the unnormalized loss distribution. Specifically, We use the following proposal distributions: 1) perform a horizontal flip with probability $0.5$; 2) scale the image by resizing by a factor of $r \sim \mathcal{N}(0, 0.5)$; 3) rotate $d$ degrees with $d \sim \mathcal{N}(0, 5)$; 4) crop $x$ or $y$ in the horizontal or vertical direction, each drawn i.i.d. $\sim \mathcal{N}(0, 2)$ and then rounded to the nearest integer value. Additionally, at each value of the annealed $t$, we perform 20 burn-in steps and only keep the last value for estimation purposes.

We compare the estimates of the functional $q$-norm of the loss over this perturbation distribution generated by the Monte Carlo estimator and by the path sampling estimator using Gaussian random-walk Metropolis sampling on a standard trained model and a data augmentation (MC $q = 1$) trained model. The results in Table 3 show that with larger $q$, the path sampling estimator produces better (higher) estimates than random sampling. For $q = 10^3$, the path sampling estimation approaches the adversarial loss over the transformation space, suggesting that this estimator, even without the advantage of Hamiltonian Monte Carlo, can effectively capture the entire spectrum of robustness. Estimates $\hat{Z}_{\mathrm{MC}}$ and $\hat{Z}_{\mathrm{PS}}$ were computed with $m = 500$. Due to the computational complexity, we only evaluate on the first 1000 CIFAR-10 test examples. The adversarial loss was approximated by averaging the maximum loss value encountered for each example during the Metropolis Hastings sampling process. Results were averaged over three repeated runs with different random seeds.

## 6   Conclusion

In this paper, we proposed a definition of intermediate-$q$ robustness that smooths the gap between robustness to random perturbations and adversarial robustness by generalizing these notions of robustness as functional $\ell_q$ norms of the loss function over the perturbation distribution. In order to evaluate intermediate-$q$ robustness in practice, we introduced an approach for approximating the high dimensional integral over the perturbation distribution that uses path sampling, an effective estimator based on MCMC sampling. We showed that across different datasets, models trained on different training objectives, and different perturbation distributions (both differentiable and non-differentiable

Table 3: Evaluations comparing the Monte Carlo estimates ($\hat{Z}_{\mathrm{MC}}$) and path sampling estimates ($\hat{Z}_{\mathrm{PS}}$) of the functional $q$-norm of the loss over non-differentiable spatial transformations on CIFAR-10. The intermediate-$q$ losses naturally interpolate between loss over random perturbations, and worst-case loss over the space of spatial perturbations. Despite the necessity of using Gaussian random walk Metropolis Hastings in place of gradient-based Hamiltonian Monte Carlo, the path sampling estimator still results in higher estimates of the desired integral than the Monte Carlo estimator for larger values of $q$.

| Train method | $\hat{Z}_{\mathrm{MC}}$ | | | | $\hat{Z}_{\mathrm{PS}}$ | | | | Adv. loss |
|---|---|---|---|---|---|---|---|---|---|
| | $q = 1$ | $q = 10$ | $q = 10^2$ | $q = 10^3$ | $q = 1$ | $q = 10$ | $q = 10^2$ | $q = 10^3$ | |
| Standard | 0.450 | 2.268 | 3.687 | 3.865 | 0.444 | 2.450 | 4.636 | 4.889 | 5.625 |
| MC $q = 1$ | 0.191 | 0.800 | 1.246 | 1.300 | 0.186 | 0.879 | 1.615 | 1.711 | 2.021 |

cases), our path sampling approach produces much better estimates of the integral than simple Monte Carlo sampling. Additionally, we illustrated the benefit of using the gradient-based Hamiltonian Monte Carlo method as the MCMC sampler when the loss is differentiable with respect to the perturbation distribution. Lastly, we highlighted the possibility of training using these estimators, and showed that models trained on MNIST using these estimators *do* exhibit improved intermediate-$q$ robustness, with path sampling again outperforming random sampling. However, the best way of estimating the integral during training of deep networks remains an open question, due to the computational complexity of accumulating enough MCMC samples at each epoch to get a reasonable estimation.

**Acknowledgements**   Leslie Rice, Anna Bair and Huan Zhang are supported by a grant from the Bosch Center for Artificial Intelligence.

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
