# OpenReview forum: "Robustness between the worst and average case"
_NeurIPS.cc/2021/Conference — NeurIPS 2021 Poster_

### Official Review · Reviewer_K1Rk · 2021-07-13

**Rating:** 6
**Confidence:** 4

**Summary:**

This paper proposes to tackle robustness both to adversarial noise and to random noise. The authors motivates the use of the Lp norm of the loss for a given noise distribution as an interpolant between robustness to adversarial attacks and random noise. The approach is well sounded.

**Limitations And Societal Impact:**

Understanding adversarial problems is a important topic, since it is a threat for ML systems in productions.

**Main Review:**

The paper is easy to read to follow. I enjoyed the HMC sampling part that I did not know before. The experiments are nice and justifies the efficiency of the HMC sampling.


Remarks:
- The recovered loss as $p\to\infty$ is not exactly the adversarial loss. The authors must be careful because it tends towards the essential supremum for the measure $\mu$ and not the maximum on the support of $\mu$. I can discuss with the authors if they are not familiar with it.
- The paper lacks a bit of rigor. "For $f$ a function" for instance is not well defined. Which space do they consider in the whole paper? Which problem they consider? What assumptions on the loss? etc. I really suggest the authors to work again on that point
- The algorithm proposed by the authors to compute the integral seems dereasonably complicated at first sight in comparison with the naive sampling: $\delta_1,\cdot \delta_n$ from $\mu$ and return $\left(\frac1n\sum l(h(x+\delta_i),y)^p\right)^{1/p}$. Although it is justified experimentally, If the estimator proposed by the authors has a smaller variance, the authors should prove it. The authors should also plot the time of computation, because one gradient computation is very costful.

Other interpolations/related work:

In the paper [1], the authors proposed an entropic regularization of the adversarial problem. One can remark this regularization can be interpreted as an interpolation between a random noise and adversarial noise too.


Rating:

For now, I rate the paper as weak reject. I really suggest the authors to work on the rigor of their paper. Please define all the objects, etc. It can only make a better paper. Moreover, I suggest the authors to give a provable justification for the use of the HMC method.


[1]Mixed Nash Equilibria in the Adversarial Examples Game. Laurent Meunier, Meyer Scetbon, Rafael Pinot, Jamal Atif, Yann Chevaleyre. ICML2021

**Time Spent Reviewing:**

2

---

> ### Author Response · Authors · 2021-08-10
> **Thank you for the thorough comments; we will improve the rigor of our paper**
>
> Thank you for your thorough comments.
>
> To start, we should emphasize that we’re indeed happy to add a bit more rigor (e.g., continuity assumptions, supremum / essential supremum distinctions, etc).  These points are largely satisfied for common neural network functions and losses, and thus we wanted to focus more on the conceptual level, but we agree there is benefit to including them, and thus we’re happy to do so.  We hope that even with these points just mentioned in the rebuttal for now, to be included in the final version, you’d consider the paper to be a valuable contribution to the current literature in robustness.
>
> Specifically, we address the following points you made in your comment.
>
> - **Essential Supremum vs Supremum:** Because in our consideration, $f$ is a neural network loss function, and is then continuous for almost every commonly used neural network, we can make the assumption that f is a continuous function. Then essential supremum is the same as supremum. We will add this assumption to the paper.
>
> - Assumptions on $f$: To be more precise, we want the p-th power of the loss function (assuming to be always positive) to be integrable with respect to its probability measure. If $\mu$ is defined as a density we want $L(h(x+\delta),y)^p \mu(\delta)$ to be Lebesgue integrable. This again should be satisfied by common choices of densities (continuous over the region of integration), neural networks, and continuous losses (note that this therefore of course does _not_ apply to 0/1 loss, just to the underlying e.g., logistic loss).
>
> - **Variance of the estimator:** We agree with this point, though because we are ultimately approximating samples via HMC (e.g., without sufficient burn-in, and for densities that may not be tractably analyzed), it is difficult to prove much about the variance actually used, and thus we rely largely on empirical evidence.  However, regarding the variance of the path sampling estimator, Gelman and Meng, 1998 compare theoretical Monte Carlo errors of importance, bridge, and path sampling estimators in Table 1 of their paper. This table is calculated using Gaussian distribution and does not exactly match our usage (we don’t have too much assumption on the distribution so variance is much harder to derive), but it provides good insights on the efficiency of the estimator. Basically, the variance for naive sampling has an exponential factor while for path sampling it is polynomial.
>
> - **Wall clock time.**  We should emphasize that even with its added complexity (which actually is relatively small compared to the time to evaluate the network), the HMC method is also substantially faster from a pure time perspective (it essentially looks no different than Figure 1).  We will add the corresponding plot to the paper.
>
> - **Additional Related Work:** Thank you for pointing us to this additional related work on entropic regularization of the adversarial problem. We will include it in our related work section and discuss its interpolation between a random noise and adversarial noise.
>
> - **A Proof for HMC:** We use HMC as a tool for sampling from the distribution defined in our problem. Generally, it is believed that HMC is a successful sampling approach especially in our setting of high dimensional inputs, and in Figure 1 and Figure 2 (in appendix) we empirically show that HMC converges better than random sampling. We believe that the main focus of our paper is to study the practically important problem of evaluating model robustness and propose a novel metric bridging the gap between random data augmentation and adversarial robustness, rather than prove certain properties of HMC. Analogously, many existing works on adversarial robustness rely on gradient descent, but it is not their focus to show gradient descent does converge.

---

> > ### Comment · Reviewer_K1Rk · 2021-08-17
> > **Thanks**
> >
> > Thank you for your response.
> >
> > While I trust the authors they will add some more background in their paper, I understand that adding some proofs to HMC might be complicated. I also think such proofs might give the paper much more impact. That said, I am fine with the answer and I am eager to improve my rating to Weak Accept (6).

---

### Official Review · Reviewer_Ausy · 2021-07-16

**Rating:** 7
**Confidence:** 4

**Summary:**

This paper proposes a definition of adversarial robustness that generalization current definitions of robustness, i.e. the loss incurred by a hypothesis when the input is contaminated by samples drawn from a perturbation set. The definition is based on a an Lp functional over the perturbation space, and interestingly it comprises the uniformly at random and adversarial cases when p=1 and p=\inty, respectively. Subsequently, in order to approximate this quantity, the authors propose a MCMC sampling approach of the partition function of an alternative density. The authors demonstrate the proposed definition numerically for MNIST and CIFAR datasets, showing that their approach performs better than a naive MC sampling. Additionally, they show that if models are trained by minimizing this new notion of p-robustness, even while approximate, they result in better robustness across this spectrum.


**Limitations And Societal Impact:**

Adequately addressed

**Main Review:**

This paper is very well written and interesting. Their definition is valuable and insightful, as it generalizes current notions of robustness and reconciles two ideas that are often treated differently: adversarial perturbations and random perturbations from some distribution (as in distribution shifts). The proposed estimator is intuitive, and the numerical results confirm their effectiveness.

My only major comment regards their reflection approach to their HMC sampling process to remain within the perturbation constraint set, which they authors explain for the case of L_\infty perturbations. Can the authors clarify if this is also applicable to general Lp balls? More generally, must this constraint set be convex for this to work? This question is motivated by the case of more general perturbations, such as those in color, contrasts, etc.

Minor comments:
- d\delta missing in Eq 11
- d\theta missing in Eq 14 and 15
- It was not immediately clear to me what the first column ("Method") in Table 1 and 2 referred to. Maybe the authors can clarify in the text, or table caption, that this refers to the way the models are trained.

**Time Spent Reviewing:**

2

---

> ### Author Response · Authors · 2021-08-10
> **Thank you for recognizing our contributions! Our approach is applicable beyond l_inf norm balls**
>
> Thank you for correctly recognizing the contributions of our paper. We really appreciate your encouraging feedback.
>
> Our definition 1 does not rely on the $\ell_\infty$ ball, and the distribution of perturbation noise is defined as a general density $\mu$. As long as $\mu$ is well defined, HMC sampling can be applied. Although we have discussed a few techniques specifically for improving the sample efficiency for the $\ell_\infty$ norm ball case where $\mu$ is a box-constrained uniform distribution (e.g., the reflection in HMC), the general principle of HMC sampling can be applied to an arbitrary perturbation set, since the perturbation set is simply the domain of the probability density to be sampled.
>
> With regards to your additional comments:
> - We will fix the missing d\delta in Eq 11 and d\theta in Eq 14 and 15 in the paper.
> - We can rename the column “Method” to “Training method” to make that more clear.

---

### Official Review · Reviewer_J5Fx · 2021-07-17

**Rating:** 5
**Confidence:** 3

**Summary:**

The paper presents a unified characterization of classifier robustness, which covers both the worst-case adversarial robustness and robustness to random perturbations. The authors mainly focus on the problem of estimating the “intermediate-p robustness” for general p. By leveraging the techniques for partition function estimation, they propose a MCMC method for estimating the intermediate-p robustness. Theoretical justifications of the consistency of the proposed estimator are provided, and empirical evaluations on MNIST and CIFAR-10 are given.

**Limitations And Societal Impact:**

The authors addressed the limitations of their work, but do not discuss the societal impact.

**Main Review:**

The unified view of classifier robustness is new and interesting to me. However, I have a hard time understanding when the intermediate-p robustness (1<p<+infinity) is useful, which is the focus of the paper. Although the paper presents a very brief argument in lines 126-129 regarding this, without specific examples or further evidence, I cannot see the significance of the considered problem. This is my major concern of the paper. If the authors can convince me the user cases of intermediate-p robustness, I would be willing to raise my score.

Other minor comments are listed below:
1.	How to interpret the results of Table 1 and Table 2? What loss function is used?

2.	How many MCMC samples are needed for producing a good estimate? In other words, what is the sample efficiency of the proposed estimator? It would be nice if the authors can show the convergence curves.

3.	For worst-case adversarial robustness, a more common evaluation metric is robust accuracy (choose 0-1 loss as the loss function in Equation 1). It would be better to provide the robust accuracy for each classifier considered.


**Time Spent Reviewing:**

2 hours

---

> ### Author Response · Authors · 2021-08-10
> **Thank you for your review! Regarding the significance of intermediate-p robustness:**
>
> Thank you for your review.
>
> We’d like to begin by addressing the most important meta point you make here, which seeks to better understand the significance of intermediate-p robustness.  This is a great point, and we’re absolutely happy to incorporate more of this broader discussion into the paper.
>
> The truth is that this is somewhat of a subjective point, because the ultimately “usefulness” of defining a new metric like this depends ultimately on what we want to accomplish with ML models in the first place.  The foundational argument we would first make is the following: as whole, as evidenced by the vast amount of work in areas, the field clearly believes that robustness to worst-case perturbations is a useful metric by which to evaluate ML models (this is ultimately the foundation for virtually all work in adversarial robustness); and likewise, there is also the clear understanding that average-case robustness is also a useful metric by which to evaluate models (in the context of “natural” robustness or e.g., data augmentation).  But the challenge has been, frankly, that these two types of robustness are typically seen as largely separate notions.  Thus, we believe there is an inherent value in generalizating these two notions in a natural fashion that places them within a broader spectrum.  And additionally, because the main criticisms of worst-case robustness is precisely that it focuses “too much” on the worst case, and likewise the criticism of average-case is that it is “not robust enough”, it seems very likely that what people “actually” often want in terms of robustness is precisely something in this middle ground.  While defining a new metric in this sense often involves some conceptual hurdles before it becomes “accepted” as a reasonable measure of performance, we fundamentally believe that better understanding the tradeoffs here (coming along, as we have done, with a method for actually computing these intermediate quantities), is quite important.
>
> With that general discussion as the primary motivation, we also emphasize some of the more immediate practical benefits of the approach: for example that intermediate-p robustness is a useful interpolation between data augmentation and adversarial training. For instance, one may wish to train a model to have better robust performance than data augmentation alone can provide, but without sacrificing too much standard performance. Intermediate-p robustness allows for the adjustment of this trade-off by tuning the value of p.
>
> There are also some technical considerations that make this a better metric: under our formulation, the perturbation can follow a Gaussian distribution with an unbounded perturbation region. Traditional adversarial robustness is not meaningful in this scenario, because the sample space is unbounded and the worst case perturbation (p=$\infty$ in our notation) can be arbitrarily far away from the original data point, leading to arbitrarily worse loss. Considering just random data augmentation from Gaussian noise (p=1 in our notation) is also insufficient because random Gaussian samples are rarely good adversarial examples. Our intermediate-p norm evaluation allows us to consider a middle ground where the model is robust under a certain degree of adversarial noise (stronger than randomly sampled Gaussian noise) while the evaluation is not hindered by very rare events (e.g., an extremely low probability Gaussian sample far away from the original input).
>
> Our response to the minor comments you listed are as follows:
>
> 1. The column labeled “Method” in Tables 1 and 2 corresponds to the training method used. The column labeled “Standard” corresponds to standard cross entropy loss. The columns labeled “RS” and “HMC” correspond to estimations of the functional lp norm of the cross entropy loss function evaluated over the perturbation distribution, where “RS” is the random sampling estimator in equation 6, and “HMC” refers to the path sampling estimator in equation 8, with samples drawn using Hamiltonian Monte Carlo. Lastly, the column labeled “PGD-100” corresponds to the PGD adversarial loss with 100 iterations. See section 5.1 for additional description of Tables 1 and 2.
> A few key observations can be made in Tables 1 and 2. First, for all models (rows) random sampling results (“RS” columns) produce much smaller estimates than HMC (“HMC” columns) for larger p. It indicates that random sampling fails to converge when p is large and more efficient sampling such as our HMC based approach is required. Second, for all models (rows), when p is larger for HMC sampling (“HMC” columns), the robustness estimates become closer to 100-step PGD (“PGD-100” column); when p=1, HMC results are identical to random sampling (“RS” columns). This indicates that our intermediate-p robustness does bridge random data augmentation (p=1) and adversarial robustness (p=$\infty$). Third, when looking at different models (rows), models trained with a larger p produce lower PGD-100 loss, indicating that training with a larger p gains more adversarial robustness.
>
> 2. The convergence of the HMC-based path sampling estimates and the random sampling estimates for p=1 and p=100 are shown in Figure 1. Additional convergence plots for p=10 and p=1000 can be found in Figure 2 in the appendix. We also plot the convergence of the path sampling estimate using the Metropolis-Hastings sampler for spatial transformations on CIFAR-10 in Figure 3 in the appendix.
>
> 3. We can include an additional column in the paper for robust accuracy. We’ve included the corresponding results below for MNIST ($\ell_\infty$, $\epsilon$=0.3, corresponding to PGD-100 evaluation in Table 1):
>
> | Training method | Robust accuracy |
> | ----------- | ----------- |
> | Standard | 4.69% |
> | RS $p=1$ | 25.21% |
> | RS $p=10$ | 43.53% |
> | RS $p=10^2$ | 47.27% |
> | RS $p=10^3$ | 45.45% |
> | HMC $p=1$ | 22.27% |
> | HMC $p=10$ | 43.57% |
> | HMC $p=10^2$ | 65.11% |
> | HMC $p=10^3$ | 69.45% |
> | PGD-50 | 91.55% |
>
> We hope our response is helpful for the reviewer to understand the motivation and importance of our work, and we will really appreciate it if the reviewer can reevaluate our paper based on our response. Thank you.

---

> ### Author Response · Authors · 2021-08-31
> **We hope the reviewer can check out our response and reevaluate our paper**
>
> Dear Reviewer J5Fx,
>
> We thank the reviewer again for your very constructive feedback. Since the discussion period is ending soon, we hope the reviewer can take a look at our response, especially on the significance of intermediate-p robustness.
>
> In short, our intermediate-p robustness is a theoretical framework that bridges the gap between the adversarial robustness (which focuses “too much” on the worst case) and the “natural” average-case robustness (e.g., data augmentation, which is “not robust enough”). Existing approaches consider the two types of robustness separately; we **unify them into a single view**, and we can **interpolate between the two**. To demonstrate a practical benefit, in our response, we showed an example with Gaussian noise perturbations where the traditional worst-case perturbation is ill-defined, while our intermediate-p robustness can still be evaluated. We believe our generalized metric of robustness can give researchers and practitioners a better understanding of robustness, and allow them to develop new methods to obtain better robust performance than data augmentation, but without sacrificing too much standard performance. We will add these discussions to our paper.
>
> We have also addressed your three specific questions in our response. We hope the reviewer now has a better picture of the problem considered in our paper and reevaluates our paper based on our response. Please kindly let us know if you have any additional comments or questions. Thank you.
>
> Sincerely,
> Paper 10671 Authors

---

> ### Author Response · Authors · 2021-09-07
> **We sincerely hope the reviewer can check out our response before the final decision is made**
>
> Dear Reviewer J5Fx,
>
> We thank you again for your helpful and insightful review. Since the main question from your review was on better understanding our proposed intermediate-p robustness metric, we will be very grateful if you could check out [our response](https://openreview.net/forum?id=Y8YqrYeFftd&noteId=QWMTj7FZ7uz) which discusses the importance and potential use cases of our metric. We would really appreciate it if you could reevaluate our paper based on our discussions before the final decision is made by the AC. Thank you.
>
> Sincerely,
> Anonymous Authors

---

### Decision · Program_Chairs · 2021-09-27

**Decision:**

Accept (Poster)

**Comment:**

The paper presents a new notion of robustness that interpolates between random perturbations and adversarial perturbations. The authors also present a MCMC based approach to evaluate the proposed robustness notion. All the reviewers agreed that this is an interesting and insightful notion of robustness and the results presented are above the bar for NeurIPS. There were concerns raised about the significance of the proposed definition, but the author response in this direction has been satisfactory. The authors should take into account the reviewers' comments before preparing the final version.